# Proteomic and Bioinformatic Profiling of Transporters in Higher Plant Mitochondria

**DOI:** 10.3390/biom10081190

**Published:** 2020-08-16

**Authors:** Ian Max Møller, R. Shyama Prasad Rao, Yuexu Jiang, Jay J. Thelen, Dong Xu

**Affiliations:** 1Department of Molecular Biology and Genetics, Aarhus University, Forsøgsvej 1, DK-4200 Slagelse, Denmark; 2Biostatistics and Bioinformatics Division, Yenepoya Research Center, Yenepoya University, Mangaluru 575018, Karnataka, India; drrsprao@gmail.com; 3Department of Electrical Engineering and Computer Science, Bond Life Sciences Center, University of Missouri, Columbia, MO 65211, USA; yjm85@mail.missouri.edu (Y.J.); xudong@missouri.edu (D.X.); 4Department of Biochemistry, University of Missouri, Bond Life Sciences Center, University of Missouri, Columbia, MO 65211, USA; thelenj@missouri.edu

**Keywords:** ABC transporter, aquaporin, ATP synthase, ion channels, mitochondrial carrier family

## Abstract

To function as a metabolic hub, plant mitochondria have to exchange a wide variety of metabolic intermediates as well as inorganic ions with the cytosol. As identified by proteomic profiling or as predicted by MU-LOC, a newly developed bioinformatics tool, *Arabidopsis thaliana* mitochondria contain 128 or 143 different transporters, respectively. The largest group is the mitochondrial carrier family, which consists of symporters and antiporters catalyzing secondary active transport of organic acids, amino acids, and nucleotides across the inner mitochondrial membrane. An impressive 97% (58 out of 60) of all the known mitochondrial carrier family members in *Arabidopsis* have been experimentally identified in isolated mitochondria. In addition to many other secondary transporters, *Arabidopsis* mitochondria contain the ATP synthase transporters, the mitochondria protein translocase complexes (responsible for protein uptake across the outer and inner membrane), ATP-binding cassette (ABC) transporters, and a number of transporters and channels responsible for allowing water and inorganic ions to move across the inner membrane driven by their transmembrane electrochemical gradient. A few mitochondrial transporters are tissue-specific, development-specific, or stress-response specific, but this is a relatively unexplored area in proteomics that merits much more attention.

## 1. Introduction

Eukaryotic multicellular organisms need to exchange energy, matter, and information between the environment and their cells, between their cells, and within their cells. To perform these tasks, they need a diverse array of specialized proteins to move ions and molecules across the biological membranes, which delimit the cells and the subcellular compartmentation. These proteins are collectively known as transporters, which include carriers, channels, and pumps [1].

The mitochondrion is a metabolic hub, not only for energy metabolism—the tricarboxylic acid (TCA) cycle and oxidative phosphorylation—but also for the biosynthesis of coenzymes, amino acids, some fatty acids, and lipids [2,3]. In photosynthetic cells in the light, there is a massive flow of fixed carbon from the chloroplasts to the rest of the cell, especially to the mitochondria via the cytosol, but metabolic cooperation between plastids and mitochondria also takes place in darkness [4,5]. Retrograde signaling from the mitochondria to the nucleus probably involves export of peptides [6,7]. In addition to this, to grow and divide the mitochondria one needs to import the vast majority of its proteins as well as some tRNAs and rRNA [8,9,10].

All of these processes require the presence of many different transporters in the mitochondria. The outer membrane contains only two types: (i) Porin also called voltage-dependent anion channel (VDAC) or voltage-dependent gated ion channel (VIC), which makes the outer mitochondrial membrane (OMM) permeable to all molecules smaller than 5 kDa, obviating the need for other transporters of small ions and molecules [11,12]. (ii) Translocase Outer Membrane (TOM), the subcomplex of the Mitochondrial Protein Translocase (MPT) responsible for importing proteins across the OMM. In addition to Translocase Inner Membrane (TIM), the MPT subcomplex responsible for importing proteins across the inner mitochondrial membrane (IMM), the IMM contains many other transporters of several different classes. The mitochondrial transportome was comprehensively reviewed by Lee and Millar [13].

It is the purpose of this review first to compile a list of the transporters identified by proteomic profiling of isolated plant mitochondria. This list will then be compared to a list of transporters predicted by MU-LOC [14], a newly developed program, to predict mitochondrial proteins based on their amino acid sequences and their gene expression patterns. Finally, for each transporter class or family, we will briefly discuss the properties of the transporters present in plant mitochondria.

## 2. The Experimental Proteome and Transportome in Plant Mitochondria

### 2.1. The Experimental Mitochondrial Proteome

The mitochondrial proteome has been characterized in some depth in *Arabidopsis thaliana* cell cultures and in potato (*Solanum tuberosum* L.) tubers. In both, almost 1100 proteins were identified as summarized by Rao et al. [8]. Since then, Senkler et al. [15] published what they called the ‘’mitochondrial complexome of *Arabidopsis thaliana*’’, in which they identified 1359 proteins involved in various complexes both in the membranes and in the soluble fraction. Altogether, the mitochondrial proteome in a plant probably contains 2000–2500 proteins [8] or about 10% of the total *Arabidopsis* proteome (Table 1). Plant mitochondrial DNA encodes at most 40 proteins [16], which are synthesized inside the mitochondria on mitochondrial ribosomes, while the remaining 2000+ proteins are encoded in the nuclear DNA, synthesized on cytosolic ribosomes, or on polysomes associated with the mitochondrial surface, and imported across the OMM and IMM [17].

The largest protein groups in the identified proteome of both *Arabidopsis* and potato mitochondria are related to energy and metabolism, with around 150 and 200 proteins, respectively; protein fate, protein synthesis, and RNA processing are each represented by approximately 100 proteins, while transport has around 50 proteins [8]. In this Gene Ontology (GO) nomenclature, many transporters are listed under different GO terms, e.g., ATP synthase subunits are found under energy. The actual number of identified transporters is therefore much larger, as discussed below. However, it is not just the number of unique proteins that is important. The abundance of each protein is also important, and Salvato et al. [18] used spectral counting to estimate the abundance of the identified proteins. The 52 proteins under the GO term Transporter showed an overall average abundance, but with significant variation among the transporters. Fuch et al. [19] went one step further by estimating the copy number of the individual proteins and protein complexes present in a single mitochondrion. Based on that, they could then calculate the surface area occupied by the various membrane proteins. They found that VDAC and TOM cover 34% and 12% of the surface area of the OMM, respectively. The five respiratory complexes cover 18% of the IMM, while the most abundant of the other carriers—ADP/ATP carrier, phosphate carrier, the uncoupling protein, and the tricarboxylate/dicarboxylate carrier—cover a total of about 11% of the IMM [19].

### 2.2. The Experimental Mitochondrial Transportome

Membrane transporters, and other transmembrane, integral-membrane proteins, are generally more difficult to identify than soluble proteins using “bottom-up” proteomics due to the paucity of charged and polar amino acids, which most sequencing-grade proteases recognize as cleavage sites. Moreover, these proteins with their hydrophobic transmembrane helices are more difficult to solubilize, and the large hydrophobic peptides formed do not ionize well for the mass spectrometry detection [20,21,22,23].

In-depth proteome profiling has only been done on mitochondria isolated from a very limited number of plant species and cell types, notably *Arabidopsis* cell cultures and potato tubers. Furthermore, the plants/cell cultures have mostly been grown under standard environmental conditions (no stress), and none of the tissues or cells have been photosynthetic. The only major exceptions to this are two developmental studies in germinating rice seeds under normoxia and hypoxia [24] and in maize embryos during seed development [25]. For that reason, the experimentally characterized mitochondrial proteome, although quite comprehensive, can be expected to lack cell-specific, tissue-specific, and environmental-specific mitochondrial proteins and protein isoforms.

In spite of these limitations, 128 transporters have been experimentally identified in *Arabidopsis* mitochondria (Table 1). The most numerous are the secondary transporters with 71 proteins, including 58 belonging to the Mitochondrial Carrier (MC) family. There are also many F-ATPases (ATP synthase subunits), and MPT proteins (subunits in the TOM and TIM complexes). Finally, 11 channels or pores have been found and most of them water channels (aquaporins) or OMM porin, as well as ion channels (Appendix A). We will discuss the different transporter classes and families in more detail after looking at the predicted mitochondrial transportome.

## 3. Mitochondrial Transporters—Predictions

We accessed the database TransportDB 2.0 [26] and extracted the membrane transport proteins in *Arabidopsis* and rice and, for comparison, in *Homo sapiens* and *Mus musculus*, similar to what was done by Hwang et al. [27]. The results are shown in Table 1. For most of the protein groups, the numbers are similar to those of Hwang et al. [27]. For plants, the only major difference is found for the group “Other ATP-dependent transporters”, where Hwang et al. [27] found 78 and 63 transporters in Arabidopsis and rice, respectively. We have divided this group into three groups—F-ATPases, MPT family, and Other ATP-dependent—but their sum in Table 1 is markedly higher for both *Arabidopsis* and rice (119 and 110 transporters, respectively). The grand total number of transporters (about 1280 proteins) makes up 5% of the total proteome in *Arabidopsis*, but only 2% in rice due to its enormous proteome (Table 1). The total number of transporters is about the same in plants as in human and mouse, but the distribution between transporter classes is quite different. Plants have twice as many ATP-binding cassette (ABC) transporters and about 200 more secondary transporters than mammals, while the humans and mice have a staggering 588 and 470 ion channels, respectively, or more than three times as many as plants. For comparison, the unicellular eukaryote *Saccharomyces cerevisiae* has only 341 transporters (~5.4% of the proteome), while *Escherichia coli* has 661 transporters (~12.3% of proteome) [26, 27, TransportDB 2.0]. The distribution of different classes of transporters in yeast and Arabidopsis is similar. On the other hand, *E. coli* has almost twice as many ABC transporters (246 or 37.2% of total transporters versus 9.7% in Arabidopsis), but far fewer other ATP-dependent class of transporters. *E. coli* also has a relatively lower proportion of secondary active transporters (281 or 42.7%) and ion channels (3.2%) compared to Arabidopsis (61.1% and 11.8%, respectively).

In the process of investigating and compiling the mitochondrial proteomes [8,18], it became clear that the existing prediction programs could at most recognize about 50% of the experimentally identified proteins as mitochondrial in spite of the fact that more than 90% of these proteins were probably bona fide mitochondrial. Transporters were particularly poorly predicted, with the best program identifying only 13 out of 59 proteins [8,18]. We, therefore, developed a new prediction program, MU-LOC, which was significantly better at predicting mitochondrial proteins over six state-of-the-art tools for plant mitochondrial targeting prediction as benchmarked on two independent datasets [14]. It was trained based on amino acid composition, protein position weight matrix, and gene co-expression information using a deep neural network, and has the advantage of predicting plant mitochondrial proteins either possessing or lacking N-terminal pre-sequences.

When all the transporters from *Arabidopsis* and rice (Appendix A) were processed by the MU-LOC program, 143 and 140 of the proteins, respectively, were predicted to localize to the mitochondria (Table 1, Appendix A), which is about 11% of the total number of transporters in *Arabidopsis*. Thus, the mitochondrial proteome, which makes up about 10% of the total proteome (see above), contains a proportional number of transporters at least in *Arabidopsis*. It is encouraging to see that the subgroups, F-ATPases, MPT family, and MC family, which are known to be predominantly mitochondrial [28,29], are heavily predicted to belong in the mitochondria (Table 1, Appendix A). The overlap between the experimental and the predicted proteins will be discussed separately below under each transporter class and family.

MU-LOC was developed to predict mitochondrial proteins in plant cells, where one of the challenges is that hundreds of proteins are dually targeted to mitochondria and plastids [30,31]. The protein features used to discriminate (see above) are actually species/kingdom neutral, so MU-LOC should also be able to predict mitochondrial localization in animal cells. We, therefore, applied MU-LOC to the total transportome in human and mouse cells (Table 1, Appendix A). The numbers of transporters of different classes predicted by MU-LOC to be mitochondrial in mammals do not differ markedly from the values for plants and certainly do not reflect the large differences in the total transportome. The high diversity in ion channels probably developed in animals in parallel with muscles and nerves. Here, rapid changes in plasma membrane potential are required and only ion channels can provide that with their short response times and large capacities [27]. It therefore makes perfect sense if there is no evolutionary increase in the number of ion channels in animal mitochondria, where the membrane potential is harnessed to produce ATP and/or transport metabolites across the IMM and where a sudden membrane potential collapse would have very negative consequences for cellular metabolism.

The ability of MU-LOC to predict mitochondrial localization of membrane transporters appears to be equally good for mammalian cells, although experimental verification is lacking. We consider this to be outside the scope of this review.

## 4. The Different Transporter Classes and Families

The *Arabidopsis* mitochondrial transportome contains 128 experimentally identified proteins and 143 predicted proteins, or 211 proteins in all, which is 17% of the total number of proteins in the *Arabidopsis* transportome. Out of the 211 proteins, 61 proteins (29%) were both predicted and experimentally identified (Figure 1).

### 4.1. ATP-Dependent Transporters

#### 4.1.1. ABC Transporters

A total of 13 ABC transporters are predicted by MU-LOC to be mitochondrial in Arabidopsis, while only six ABC transporters have been experimentally found to date (Table 1, Appendix A). Three proteins were both predicted and found: iron-sulfur clusters transporter ATM1 (At4g28630), ATM2 (At4g28620), and ATM3 (At5g58270). They all take part in the export of iron-sulfur clusters synthesized in the matrix to the intermembrane space from where the clusters are distributed to the rest of the cell [32,33].

In fact, iron-storing ferritin and the entire iron-sulfur biosynthesis pathway were found in potato mitochondria: frataxin, iron-sulfur cluster assembly proteins, iron-sulfur cluster scaffold protein, iron-sulfur cluster co-chaperone protein, Cys desulfurase, and ferredoxin [18,34].

#### 4.1.2. The Mitochondrial Protein Translocator (MPT) Family

The MPT family consists of subunits in the TOM and TIM complexes, where the majority of the subunits have been identified and about half predicted (Table 1, Appendix A) [35]. This transporter family is the subject of another review in this Special Issue and will not be treated any further here.

#### 4.1.3. F-ATPases

The *Arabidopsis* genome encodes 44 F-ATPase transporters belonging to the ATP synthase in mitochondria and plastids as well as to the P-type ATPase in the vacuole. Of these, MU-LOC predicted 17 to be mitochondrial and 16 were found experimentally, while the overlap was about 50% (9 proteins).

#### 4.1.4. P-ATPases

The *Arabidopsis* genome encodes 50 P-ATPase transporters belonging mainly to ATPases in the plasma membrane. Of these, MU-LOC predicted four to be mitochondrial, and two were found experimentally with no overlap. The experimentally found were At1g54280, a phospholipid transporting ATPase annotated to be located in the plasma membrane, and At4g33520, a copper-transporting ATPase annotated to be located in the plastid envelope (as annotated in UniProt). Both would be very useful to have in the IMM.

### 4.2. Secondary Transporters

This is by far the largest group of transporters in plants and in plant mitochondria (Table 1, Appendix A), where they are located in the IMM. These transporters are either symporters or antiporters, and use the electrochemical gradient established across the IMM to move ions into or out of the mitochondrial matrix. The size and diversity of transporters in this family reflect the metabolic complexity of mitochondria and its importance as an energetic conduit for the cell [13,28,29].

#### 4.2.1. The Mitochondrial Carrier (MC) Family

The MC family (MCF) is predicted to contain 58–60 members in *Arabidopsis* [28, TransportDB 2.0] and 50–61 members in rice [24, TransportDB 2.0]. The majority of the MCF members are localized in the IMM, although at least 10 are found in plastids, peroxisomes, endoplasmic reticulum, and plasma membrane [29].

About half of all the MCF members are predicted to be found in the mitochondria of both *Arabidopsis* (30 out of 60) and rice (33 out of 61) (Appendix A). In *Arabidopsis*, 58 members have been experimentally identified (including all the predicted), in other words, a very satisfying 97%, while in rice 49 have been identified (including 29 of the predicted), producing an 80% ‘’recovery’’ rate (Appendix A, Figure 1). The almost 100% recovery rate for MCF members in proteomic profiling of isolated mitochondria means that most, if not all, of the MC members localized to other membranes [13,29] have also been found in mitochondria. To what extent this is due to dual localization or to contamination of the mitochondrial preparations is an open question.

The MC members catalyze the transport of numerous metabolic intermediates across the IMM—nucleotides like ADP, ATP, and NAD^+^; coenzymes like coenzyme A, thiamine pyrophosphate, and folate; di- and tri-carboxylic acids like malate, oxoglutarate, succinate, fumarate, and citrate; amino acids like glutamate, aspartate, ornithine, citrulline, and carnitine; as well as phosphate and protons. All these carriers have been admirably reviewed by Palmieri et al. [28] and Toleco et al. [36], so we will not discuss them any further except briefly in connection with one important metabolite, ascorbate.

Ascorbate is very important in mitochondrial metabolism where it participates in the ascorbate-glutathione cycle that removes H_2_O_2_ produced by the respiratory chain e.g., [37]. The last step in ascorbate biosynthesis takes place on the outer surface of the IMM [38], but to date no ascorbate transporter has been identified in the IMM. Ascorbate transport was measured in plant and rat liver mitochondria by Scalera et al. [39]. The protein responsible was in the size range 28–35 kDa, which is where most of the MC family transporters are found, but the gene was not identified. An ascorbate transporter has been identified in the chloroplast envelope [40], which could in principle be dually targeted to IMM. However, since it is 60 kDa it could not be responsible for the transport activity observed by Scalera et al. [39].

#### 4.2.2. Transport of Inorganic Ions

Ca^2+^—*Arabidopsis* contains a Ca^2+^-cation antiporter family with 13 members out of which one is predicted to be mitochondrial, none has been identified by proteomics. Rice contains a Ca^2+^-cation antiporter family with 16 members, out of which three are predicted to be mitochondrial; none has been identified in proteomics. One of the predicted antiporters may be involved in Ca^2+^/Na^+^ exchange.

K^+^—*Arabidopsis* mitochondria contain three members of the Monovalent Cation: Proton Antiporter-2 (CPA2) Family. Two were found experimentally—At1g01790 K^+^ efflux antiporter 1, chloroplastic and At2g28180Cation/H^+^ antiporter 8—and one was predicted (At5g41610 cation-H^+^ antiporter (Appendix A, UniProt)). *Arabidopsis* mitochondria also contain three members of The Monovalent Cation: Proton Antiporter-2 (CPA2) Family. At5g41610 (Cation/H^+^ antiporter 18) was predicted but not found, while At1g01790 (K^+^ efflux antiporter 1, chloroplastic) was found but not predicted. All of these transporters could help the mitochondria regulate the K^+^ concentration in the matrix.

Metal ions—two members of the Cation Diffusion Facilitator (CDF) Family in *Arabidopsis* have been either found and predicted (At1g51610—Metal tolerance protein C4, tonoplast) or just predicted (At2g47830—predicted Metal tolerance protein C1, tonoplast) (Appendix A). Unless these are merely contaminants in the isolated mitochondria, they could provide the mitochondria with the means of transporting metal ions, like Pb^2+^, Ni^2+^, and Fe^3+^, across the IMM.

Sulfate—there are 14 members of the sulfate permease family in *Arabidopsis*, but only one (At1g80310) is predicted to be mitochondrial by MU-LOC, and none was found by proteomics (Appendix A). The predicted mitochondrial transporter turns out to be a molybdenum transporter located in the tonoplast (as annotated in UniProt).

Nitrite—nitrite is known to reach the mitochondrial matrix, where it can act as an alternative electron acceptor under hypoxia/anoxia [41]. No nitrite carriers are listed in Appendix A, but the nitrite carrier found in the inner envelope in chloroplasts (At1g68570) is a member of the proton-dependent oligopeptide transporter family [42]. This family has 18 members in *Arabidopsis*, and several of them transport nitrate. However, none of them is predicted to be mitochondrial by MU-LOC, and none has been found in proteomic studies (Appendix A).

#### 4.3. (Ion) Channels

##### 4.3.1. Porin (VDAC or VIC)

The VDAC/VIC family has 36 members in *Arabidopsis* (as listed in TransportDB), and only one has been predicted to be mitochondrial (Appendix A). However, six VICs have been found in *Arabidopsis* mitochondria experimentally (Appendix A), and they are all bona fide VDACs according to UniProt, but they are not listed in TransportDB (Appendix A). The OMM in isolated mitochondria behaves as if it is freely permeable to small molecules, so the VDACs do not appear to be actively gated, but it is possible that porin is involved in tRNA transport [11,12,42].

##### 4.3.2. Aquaporin

Aquaporins allow water, but also other small neutral molecules like hydrogen peroxide and ammonia, to pass membranes in both directions going from higher to lower osmotic potential [43,44]. Contrary to the conclusion by Maurel et al. [44] that “plant mitochondria seem to be deprived of aquaporins”, MU-LOC predicted that 7 out of 39 *Arabidopsis* aquaporins localized to the mitochondria, and proteomic profiling of *Arabidopsis* and potato mitochondria found six aquaporins (only one overlaps between prediction and experimental) (Appendix A). It has long been known that (plant) mitochondria can swell and shrink. For instance, the mitochondria swell rapidly when the osmotic potential decreases in the matrix due to a rapid influx of osmolytes e.g., [45], which would only be possible if aquaporins are present in the IMM.

The final steps in substrate oxidation take place in the TCA cycle in the mitochondrial matrix where glycolytic end products are oxidized to CO2. One still unsolved question is how the CO2 leaves the mitochondria and in what form? Animal cells have Na+/bicarbonate symporters and chloride/bicarbonate antiporters, but none of them are predicted to be mitochondrial by MU-LOC (Appendix A). So how does CO2 get out of the mitochondria? One intriguing possibility is through aquaporins just like other small uncharged molecules [44].

##### 4.3.3. Ion Channels

The Small Conductance Mechanosensitive Ion Channel (MscS) Family contains 11 members in *Arabidopsis,* out of which At4g00290 (mechanosensitive ion channel protein 1, mitochondrial) was both predicted and found, whereas At1g49260 was only predicted. It is possible that the mechanosensitive ion channel protein 1 can catalyze ATP-dependent K^+^ channel activity by teaming up with one of the ABC transporters [46].

Two subunits of the calcium uniporter protein [47,48]—MCU1 (At4g36820) and MCU2 (At2g23790)—have been identified in plant mitochondria (Appendix A), but they do not appear in the TransportDB list (Appendix A).

The list of *Arabidopsis* ion channels contains eight so-called chloride intracellular channel (CLIC) homologs, out of which three are predicted to be mitochondrial (At1g19570, At5g16710, and At5g36270), and one is both predicted and experimental (At1g75270). They are all annotated as glutathione S-transferases with glutathione-dependent dehydroascorbate reductase activity and appear to be able to transfer Cl- across membranes [49], possibly by a mechanism involving a redox reaction (as annotated in UniProt).

## 5. Posttranslational Modifications of Transporters

Posttranslational modifications (PTMs) of proteins are a way of regulating cellular and mitochondrial metabolism [37]. In shotgun proteomics, many mitochondrial transporters have been observed to have PTMs of various types: (i) oxidations to give carbonylated side chains on primarily Lys and Pro or sulfoxide on Met [18], although the latter is probably an analytical artefact in some cases; (ii) phosphorylation of Ser, Thr, and Tyr [50,51,52]; (iii) acetylation of Lys [53]; and (iv) conjugation of Lys side chains with oxidative degradation products of polyunsaturated fatty acids, for instance, 4-hydroxynonenal (HNE) [54,55]. Many of these modifications are no doubt regulatory, while others are damaging and lead to proteolytic degradation [37,56,57].

The modified transporters include the ATP, ADP-translocase, the phosphate transporter, and the dicarboxylate carrier; the former two are both oxidized and acetylated on multiple sites [18,53]. The effect of the modifications is so far unknown. Plasma membrane aquaporins are known to be gated by phosphorylation [44], but aquaporin phosphorylation has never been reported for plant mitochondria.

## 6. Physiological Changes in the Mitochondrial Transporters

There has been a number of studies of individual mitochondrial transporters during development or during the stress response, e.g., the Ca^2+^ uniporter [48], but very few where the whole mitochondrial transportome has been considered.

### 6.1. Changes in Rice during Development

Taylor et al. [24] studied members of the MC family in rice. The expression of 44 of the 50 MC genes in rice was quantified in different tissues during the time course of aerobic and anaerobic seed germination, during seedling growth and in response to fungal infection. Several tissue-specific carriers were identified in the shoots and flowers, and the expression of specific ATP/ADP transporters, succinate/fumarate carriers, and a dicarboxylate/tricarboxylate carrier (DTC) increased during fungal infection.

Taylor et al. [24] also used a targeted proteomic approach especially suited for identifying integral-membrane proteins [20], and succeeded in identifying and quantifying five different MC proteins in dry seeds and in seeds incubated under normoxia and anoxia. Significant differences in protein abundance between the treatments were observed for the phosphate carrier and the basic amino acid carrier.

### 6.2. Changes in Maize during Development

Wang et al. [25] studied mitochondrial development in developing maize seed embryos. They purified mitochondria from 300–600 excised embryos (2–15 g fresh weight) at five time points during the period 14–70 days after pollination (DAP) and did proteome profiling by a shot-gun method. Altogether, they identified and quantified 931 mitochondrial proteins (including 31 transporters—Appendix A) and observed that the abundance of 286 proteins changed more than two-fold during embryo development. The abundance of most of these peaked early during development either on day 14 (the first day investigated) or on day 21 after pollination. Out of the differentially abundant proteins, 11 were transporters, and 7 of these, including two DTCs, showed a pattern where the abundance peaked on day 21, i.e., quite early in development.

## 7. Conclusions

Plant mitochondria contain more than 100 transporters of all classes and families, both as determined by proteomics of isolated mitochondria and by prediction using the MU-LOC bioinformatics tool with a good overlap between the two methods. By far the largest group is the secondary transporter family Mitochondrial Carriers, which includes many of the well-known inner mitochondrial membrane carriers of organic acids, amino acids, etc. Other well-represented families are the F-ATPases and the Mitochondrial Protein Translocators, but plant mitochondria also contain ABC transporters, water channels, and ion channels. There is a real need for targeted proteomic profiling of mitochondria from a broader tissue spectrum, especially green tissues, but targeted profiling of roots and flowers would also help to obtain a more complete picture of the mitochondrial transportome in plants. It would also be useful to study the transportome under a variety of environmental conditions to identify stress-responsive transporters.

## Figures and Tables

**Figure 1 biomolecules-10-01190-f001:**
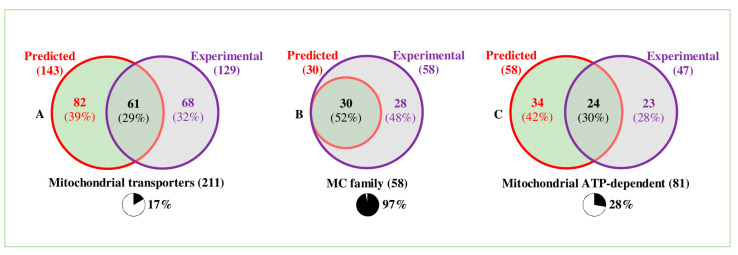
*Arabidopsis* mitochondrial transporters in Venn diagrams. (**A**) A total of 211 transporters (17% of all the transporters in *Arabidopsis*) are either predicted to be or experimentally known to be mitochondrial. Out of these, 61 are both predicted and experimental. (**B**) Nearly all (97%) Mitochondrial Carrier (MC) family members have been experimentally identified in mitochondria, while only 50% are predicted to be mitochondrial. (**C**) A total of 81 ATP-dependent transporters (28% of all ATP-dependent transporters in *Arabidopsis*) are predicted to be or experimentally known to be mitochondrial. Out of these, 24 (30%) are both predicted and experimental.

**Table 1 biomolecules-10-01190-t001:** Number of proteins in different transporter classes and families, and their experimental (proteomic) identification in isolated mitochondria of Arabidopsis and rice and mitochondrial localization prediction status (using MU-LOC) in Arabidopsis, rice, human, and mouse.

	*A. thaliana*					*O. sativa*					*H. sapiens*			*M. musculus*		
	TransportDB ^#^	Experimental *	%	Predicted ^#^	%	TransportDB ^#^	Experimental *	%	Predicted ^#^	%	TransportDB ^#^	Predicted ^#^	%	TransportDB ^#^	Predicted ^#^	%
Whole proteome	26,091	**-**	-	**-**	-	55,890	**-**	-	**-**	-	37,742	**-**	-	34,966	**-**	-
**ATP-dependent**																
ABC transporters	124	**6**	5	**13**	11	129	**0**	0	**11**	9	77	**10**	13	65	**5**	8
F-ATPase	44	**16**	36	**17**	39	50	**7**	14	**13**	26	60	**25**	42	58	**19**	33
MPT family	25	**19**	76	**10**	40	30	**2**	7	**10**	33	29	**6**	21	40	**6**	15
P-type ATPases	50	**2**	4	**4**	8	45	**0**	0	**3**	7	68	**6**	9	50	**0**	0
Other ATP-dependent	50	**4**	8	**14**	28	30	**0**	0	**10**	33	19	**4**	21	34	**4**	12
**Secondary transporters**																
MC family	60	**58**	97	**30**	50	61	**50**	82	**33**	54	63	**25**	40	64	**21**	33
Other families	722	**13**	2	**34**	5	799	**0**	0	**50**	6	511	**39**	8	473	**28**	6
**Ion channels**	151	**11**	7	**15**	10	137	**0**	0	**10**	7	588	**9**	2	470	**8**	2
**Unclassified**	53	**0**	0	**6**	11	4	**0**	0	**0**	0	52	**5**	10	36	**2**	6
Total (Transporters)	1279	**129**	10	**143**	11	1285	**59**	5	**140**	11	1467	**129**	9	1290	**93**	7

# See Appendix A for a full list of transporters from TransportDB (http://www.membranetransport.org/transportDB2/index.html) and their mitochondrial prediction status (using MU-LOC). * See Appendix A for a list of experimentally identified mitochondrial transporters.

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
