# Peer review of "Proteomic and Bioinformatic Profiling of Transporters in Higher Plant Mitochondria"

_biomolecules, 2020, doi:10.3390/biom10081190_

Round 1

Reviewer 1 Report

This is a review about transporters in plant mitochondria. It is an exhaustive compilation study very useful in this field. However, my main concern is related to the lack of a clear objective for the manuscript. It is unclear if authors want to just review plant mitochondria transporters or compare their predictive software with proteomic profiling predictions. Furthermore, conclusions (358-367) do not answer the purpose of the study indicated in lines 56-60 and sections 5 and 6 seem to be out of the scope of the manuscript.

Other points to be considered are:

-Table 1: In order to clarify, in the second row, substitute the term “predicted” for “predicted in mitochondria”. The same for “experimental”. More information about TransportDB should be provided. Comparison with transporters in bacteria would give additional relevant information.

-Table S1: Please, separate transporters by species in different Excel sheets. Check title, some species are missing.

-Lines 121-123: Please check meaning: “[27].Only for their group Other ATP-dependent transporters, which we have divided into three groups – F-ATPases, MPT family, and Other ATP-dependent – did they have a markedly higher number for both Arabidopsis and rice.”

-Line 200-202: please, check:” This transporter family is the subject of another review in this Special Issue (reference) and will not be treated any further here”.

-Lines 246-250: Regarding this author´s comment: ”The protein responsible was in the size range 28-35 kDa, which is where most of the MC family transporters are found, but the gene was not identified. An ascorbate transporter has been identified in the chloroplast envelope [40], which could in principle be dually targeted to IMM. However, since it is 60 kDa it could not be responsible for the transport activity observed by Scalera et al. [39].” Have the authors considered the possibility of a dimerization of the transporter by Scalera et al (28-35 kDa), giving to the  transporter identified in the chloroplast (60 kDa)?” 

-Line 272: What does “UniProt” mean?

Typographic errors:

-Line 69 “most40 proteins”

-Line 127: “Plantshave”

-Line 130: “[8,18],it”

-Line 134: “[8, 18].We”

-Line 167: “contains128”

-Line 289: “membranesin”

-Line 304: “Familycontains”

Reviewer 2 Report

The article is very interesting. This is an area of considerable interest, but has also been the subject of a number of other recent articles. There is therefore a lot of previous, and recent, information currently in the literature. However there are some information that should be clarified in the manuscript (major revisions) in order to be accepted for publication.

The title should be shortened is definitely too long. 

I would compare proteomics to other eukaryotes such as yeasts. Add a few sentences. This will enrich and interest the readers.

It is worth writing a similar introduction that will introduce the reader to the topic:

Proteomics is defined as the study of the proteome, or protein component, encoded by the genome. The term "proteomics" was first formulated and used by Marc Wilkins in 1994. The main goal of proteomic research is to isolate proteins produced by cells and organs, both in physiological and and pathological ones, studying their interrelationships and learning about their three-dimensional structures.

The life of a cell is based on many interrelated dynamic processes that
affect its growth and ability to reproduce and experiences. The quantity and quality of proteins in a cell is controlled not only by the rate of their biosynthesis and degradation, but also by specific processes such as post-translational modifications that modulate molecular interactions affect the stability of proteins and their localization in individual intercellular compartments - Therefore, I believe that the authors should make a summary table of all known transport proteins in plants.

It is worth mentioning biomarkers, i.e. biological features of a molecular nature, which can be used as indicators of physiological or disease processes taking place in the plant organism or as indicators for assessing the degree of response to various stress conditions.

The main analyzes used in proteomics research need to be added to the manuscript.

The authors should also add short information about biological bases, for example UniProt (Universal Protein Resource), which is a comprehensive and easily available tool containing information on protein sequences and their functions, collected on the basis of scientific literature, and NCBI (National Center for Biotechnology Information), which is a collection of several different NCBI includes, for example GenBank, i.e. a database containing a collection of genes nucleotide sequences and SwissProt, na which consists of a significant amount of information on protein functions, structure and modification post-translational as well as many other data.

I missed the disadvantages and limitations of proteomic research. Complete this information. Perhaps it is worth describing the strategies and participating transporters in the collection of various elements from the soil. The authors do not mention this at all.

Round 2

Reviewer 1 Report

Although authors have attended some of the suggestions, there are some of them that have been rejected without a relevant reason. For example, the suggested comparison with bacteria and yeast (suggested by the other reviewer) transporters should be considered. Adding some comparative information with other organisms would improve the manuscript.
